# Dynamic Asynchronous Anti Poisoning Federated Deep Learning with Blockchain-Based Reputation-Aware Solutions

**DOI:** 10.3390/s22020684

**Published:** 2022-01-17

**Authors:** Zunming Chen, Hongyan Cui, Ensen Wu, Xi Yu

**Affiliations:** 1State Key Laboratory of Networking and Switching Technology, Beijing University of Posts and Telecommunications, Beijing 100876, China; czm@bupt.edu.cn; 2School of Information and Communication Engineering, Beijing University of Posts and Telecommunications, Beijing 100876, China; wuensen@bupt.edu.cn (E.W.); YUSY@bupt.edu.cn (X.Y.)

**Keywords:** federated machine learning, security, privacy-preserving, asynchronous, poisoning attack

## Abstract

As promising privacy-preserving machine learning technology, federated learning enables multiple clients to train the joint global model via sharing model parameters. However, inefficiency and vulnerability to poisoning attacks significantly reduce federated learning performance. To solve the aforementioned issues, we propose a dynamic asynchronous anti poisoning federated deep learning framework to pursue both efficiency and security. This paper proposes a lightweight dynamic asynchronous algorithm considering the averaging frequency control and parameter selection for federated learning to speed up model averaging and improve efficiency, which enables federated learning to adaptively remove the stragglers with low computing power, bad channel conditions, or anomalous parameters. In addition, a novel local reliability mutual evaluation mechanism is presented to enhance the security of poisoning attacks, which enables federated learning to detect the anomalous parameter of poisoning attacks and adjust the weight proportion of in model aggregation based on evaluation score. The experiment results on three datasets illustrate that our design can reduce the training time by 30% and is robust to the representative poisoning attacks significantly, confirming the applicability of our scheme.

## 1. Introduction

Around the world, there are about 10 billion Internet of Things (IoT) devices with increasingly advanced computing, communication, and sensors capabilities currently [1]. Coupled with the rapid development of deep learning, it opens up endless possibilities for many applications, such as in vehicular networks and for industrial purposes. Traditional cloud-centric machine learning methods require the personal information to be stored in the data center or cloud-based server to perform model training, which results in communication inefficiency and unacceptable latency. Therefore, Mobile Edge Computing is proposed to bring intelligence close to edge networks. However, machine learning at the edge still needs to share personal information with external parties such as edge servers. Recently, Federated Learning (FL), a promising privacy-preserving technology, is proposed to address the challenges of growing privacy concerns and stricter privacy legislation. FL enables multiple clients to train the joint global model via sharing model parameters instead of raw data. However, there are two limitations to reducing federated learning performance, including inefficiency and vulnerability to poisoning attacks.

The first limitation is inefficiency. The existing federated learning parameter aggregation approach includes synchronous algorithms and asynchronous algorithms. Synchronous federated learning is inefficient because of the delay in waiting for all clients to complete training before updating the global model. This paper proposes a dynamic asynchronous algorithm considering the averaging frequency control and parameter selection for federated learning to speed up model averaging. The proposed algorithm enables federated learning to adaptively remove the stragglers with low computing power, bad channel conditions.

Another limitation is vulnerability to poisoning attacks. A poisoning attack means that malicious clients can generate anomalous parameters to mislead the model decision. Particularly, recent research works have illustrated that poisoned parameters can mislead the federated learning model on the attacker-chosen poison subtask while working well on the main task [2]. The limitations including inefficiency and vulnerability to poisoning attacks of federated learning significantly reduce federated learning performance, which motivates us to solve these problems. This paper proposes a novel local reliability mutual evaluation mechanism to enhance the security of poisoning attacks, where each parameter is evaluated over the local data of other parties. According to the evaluation scores, the server can adjust the weight proportion of model aggregation. The local reliability mutual evaluation mechanism uses the local reliability to detect poisoned parameters instead of statistical difference analysis enabling the approach to work well in the case of small data samples. The main contributions of this paper are as the following: We propose a dynamic asynchronous anti poisoning federated deep learning framework to pursue both the efficiency and security of defending against poisoning attacks. In particular, the dynamic asynchronous algorithm considering the averaging frequency control and parameter selection for federated learning is proposed to speed up model averaging. The proposed algorithm enables federated learning to adaptively remove the stragglers with low computing power, bad channel conditions.A novel local reliability mutual evaluation mechanism is presented to enhance the security of poisoning attacks. The proposed mechanism enables federated learning to detect the anomalous parameter of poisoning attacks and adjust the weight proportion of model aggregation based on the evaluation result.The experiment results on three datasets illustrate that our design can reduce the training time by 30% and is robust to representative poisoning attacks significantly compared with other state-of-art methods, confirming the applicability of our scheme.

The remainder of this paper is organized as the following. In Section 3, we provide the necessary background about federated deep learning and blockchain. Section 4 presents the details of RAPFDL, a framework for dynamic asynchronous anti poisoning federated deep learning, and analyzes its security. In Section 5, we illustrate the implementation details of RPPFDL, followed by performance evaluation in Section 6. Finally, Section 7 concludes the paper with some future research directions.

## 2. Related Work

### 2.1. Efficient Federated Learning

The existing federated learning parameter aggregation approach includes synchronous algorithms and asynchronous algorithms. Synchronous federated learning is inefficient because of the delay in waiting for all clients to complete training before updating the global model. For example, Mahan et al. proposed FedAvg [3] to perform local model training in clients, wait for all clients to complete training, and parameter uploading to aggregate the global model. To solve the problem, asynchronous algorithms have been proposed. Meng et al. proposed FedAsync [4] where the model parameter from clients will be immediately aggregated by a central server, and periodically the latest global model is sent to clients. Compared to synchronous algorithms, asynchronous algorithms is more efficient. However, there is a version gap problem of the asynchronous algorithm. Owing to federated learning clients are heterogeneous with various resource constraints. Some straggler clients with low computing power or bad channel condition still train the old version global model while the global model has been updated with model parameters from other fast clients. To solve the version gap problem, Rangwala et al. proposed the penalizing strategy [5] to improve the local learning rate of slow nodes and reduce the training learning rate of fast nodes. Jaehyun et al. proposed the strategy of allocating transmit power [6] to allocate more power to bad channel condition nodes carrying huge data and reduce the transmit power of good channel condition nodes carrying small data. Sanmi et al. proposed the weight of version gap [7], when the version gap between the local model of client and federated global model is small, the weight of version gap should be increased, and when the version gap is large, the weight of version gap should be reduced. The above works alleviate the problem of the version gap but fail to solve the problem of data volume skewing and model bias convergence. Slow nodes may have higher uploading delay due to large data volume, but their contribution is relatively higher. Frequent uploading parameters of fast nodes lead to the deviation of federated models from the convergence direction of models on other slow nodes.

### 2.2. Defenses against Poisoning Attack on Federated Learning

To solve the poisoning attacks problem, there are three approaches including cluster detection, distance detection, and similarity detection. Tople et al. has proposed Auror [8] which detects the anomalous parameter via clustering algorithm to measure the difference in the distribution of benign and anomalous parameter if malicious clients upload poisoned parameters continually. Blanchard et al. has proposed Krum [9] where parameters far from an average distance are removed to improve the robustness to poisoning attacks. However, these two methods cannot defend against the single-round attack and are not suitable for different data distribution scenarios. Recently, Yoon et al. has proposed FoolsGold [10] where anomalous parameters from malicious clients are identified via calculating similarities between various parameters from clients to defend against the poisoning attacks. Yet, it is hard to defend the attack from a single malicious client.

## 3. Background

The dynamic asynchronous anti poisoning federated deep learning is based on two key technology including federated deep learning and blockchain. This section provides necessary background knowledge about them.

### 3.1. Federated Deep Learning

A traditional deep learning model usually includes the input layer, one or more hidden layers, and the output layer. Each hidden layer has a certain number of neurons, which have multiple inputs and a single output [11,12]. Neurons between two adjacent layers are connected by weight parameters w. Each neuron i also has a bias b_i. The model parameters (denoted as (w,b)) require to be learned during model training. The goal of the deep learning algorithm is to find the optimal model parameters which reflect the relationship between the inputs and outputs so as to obtain the model which takes a sample X in the forms of a matrix or vector consisting of a real number as input and outputs the predicted result $y^{\prime }$ close to its real labels. In order to find optimal parameters, the training algorithms try to minimize the loss function $L(D,\omega )=\frac{1}{\left|D\right|}\sum_{D^{i}\in D}L(D,\omega )$, Where D represents the training set, and w is the model parameters.

Stochastic Gradient Descent is a famous approach to minimize the loss function, which iteratively finds the optimal parameters, i.e.,
(1)$$\omega^{j+1}=\omega^{j}-\eta \nabla L(D^{j},\omega^{j})$$
where $w^{j}$ represents the model parameters after the *j*-th iteration. $\nabla L(D^{j},\omega^{j})$ denotes the partial derivative of the parameter $w^{j}$ on the set $D^{j}$. η corresponds to the learning rate which controls the step of gradient descending. Because loss function is used for estimating the gap between model output and the target value, Equation (1) is able to be divided into two phases including forward propagation which calculates the predicted result of input X, and backward propagation which aims to calculate the parameter update.

In federated learning, assuming that each user n∈N has a local data set $D^{n}$. To run the stochastic gradient descent algorithm, each user n computes the local gradients via $x_{n}^{j}=|D_{n}^{j}|\nabla L(D_{n}^{j},\omega^{j})$ at *j*-th iteration [13,14]. Thus, the loss gradient can be rewritten as below,
(2)$$\nabla L(D^{j},\omega^{j})=\frac{1}{\left|D^{j}\right|}\sum_{n\in N}x_{n}^{j}$$

Then, each user can update its local model with $w^{j+1}$ returned by the cloud server. The training process repeats until the model output reaches the predetermined pre-specified precision thresholds.

### 3.2. Blockchain Technology

Blockchain is the first technology that has drawn a great deal of attention in both the academic community and industry [15,16], which is a promising technology as a decentralized, reliable, time-order ledger. The key fundamental concepts characterizing blockchain include distributed ledger, consensus algorithms, smart contracts, and cryptography. Blockchain is a distributed ledger composed of devices or nodes. Each node has a copy of the ledger, and each copy is updated individually. Nodes vote on every update on blockchain to agree on the same conclusion, and the modifications are recorded by the nodes. In a blockchain, an entity called a worker generates blocks and links them one by one to build an immutable ledger. Each block includes multiple transactions created by participants, which can record the data for a specific application. Sophisticated consensus mechanisms such as the proof of work (PoW) and the practical byzantine fault tolerance (PBFT) are needed to ensure the trans-actions consistency in the new block. Blockchain is not controlled by one entity and does not have a point where something can fail, making this technology safe and robust. The blockchain can be divided into two types, namely permissioned blockchain and permissionless blockchain. In the permissionless blockchain like Ethereum, parties can join and leave easily. In the permissioned blockchain such as Hyperledger Fabric, the set of parties are predefined and parties require permissions to join or leave. In addition, pro-developing Blockchain technologies introducing smart contract support Turing-complete programmabilities, such as Ethereum and Hyperledger. We adopt the Hyperledger blockchain platform as our experimental testbed.

Smart contracts of the blockchain, are the executable programs that enable running programs on the blockchain platform. It can get credible results by leveraging the security guarantee of blockchain. Currently, there are numerous pro-developing blockchain platforms supporting smart contracts (e.g., Hyperledger Fabric, Ethereum). Smart contracts deployed on the blockchain will be executed automatically once invoked by participants, enabling participants to reach a consensus (e.g., incentives mechanism, selection of optimal parameters updates) without depending on the centralized servers vulnerable to powerful adversaries. In a nutshell, blockchain can offer the following properties to RAPFDL, including trusted records and reliability evaluation.

Distributed ledger technologies are the digital approach for registering transactions and other similar data in multiple nodes with different locations simultaneously. There is no central administration component in distributed ledgers, thus it will not suffer from single-point failure, making this technology safe and robust. Several distributed ledger technologies are available, including Blockchain and Tangle technologies. Blockchain is a promising technology as a decentralized, reliable, time-order ledger. Tangle technologies are essentially a directed acyclic graph that can hold transactions. By running nodes in Tangle, decentralization is developed. Blockchain can be categorized into three types for use cases, namely public blockchain, consortium blockchain, and private blockchain. The comparison of three types of blockchain is listed in Table 1. A public blockchain is open to the world. The consortium blockchain is restricted, where parties require permission to join or leave. And it could be applied to many business applications. There are several consortium blockchains available such as Hyperledger, Ethereum, and Corda. As for private blockchain, it suits the centralized cases pursuing efficiency and auditability.

## 4. The RAPFDL Framework

This section presents the dynamic asynchronous and anti poisoning federated learning (RAPFDL) framework in technical details as follows: reliability evaluation, blockchain decentralized architecture, local model training, federated model aggregation.

Before introducing RAPFDL, we define the related components and concepts in RAPFDL.

Reliability evaluation: One of our key insights is delegating anomaly detection tasks to clients which is able to detect anomalous parameters via evaluating the local reliability of model parameter updates with its private dataset. Put simply, the central server sends model parameters updates to the selected clients which evaluate the accuracy performance of the parameter updates with local data according to the predefined rule. The server can adjust the weight proportion of model aggregation based on the matrix of received evaluation scores.Blockchain decentralized architecture: Most existing FL frameworks rely on a central server to aggregate model parameter updates of parties involved in FL. Compared with centralized architectures, RAPFDL inherits the blockchain architecture, which enables every party to remain modular when interacting with other parties. Rather than ceding control to central servers, every party maintains full control of private data. In addition, blockchain enables federated learning with the native ability to coordinate the entry and exit of parties automatically, further guaranteeing the auditability of the training process of the FL. Robustness can also benefit from the blockchain because of no single point of failure.Local model training: Every party performs local model training independently. After completing the training process, the party generates a contract to trade its local model parameter updates by attaching its local model parameters to the contract.Federated model aggregation: Parties of a cooperative group train a deep learning model collaboratively. The model is trained in an iterative manner after deciding on the same deep learning model and parameter initialization. All parties trade their parameter updates, and workers download the contracts to process the parameter updates in each iteration. The processed parameter updates are then sent out via smart contract namely the processing contract of blockchain. The correctly processed parameter updates are used to update the federated global model by the leader selected from workers. Each party downloads the federated model to update its local model accordingly. After that, the next iteration of federated learning begins.

RAPFDL combines federated learning and blockchain techniques to achieve dynamic asynchronous anti poisoning federated learning. Figure 1 presents an overview of the RAPFDL scheme. Assuming that parties might be unwilling to share raw data when joint training the global model without the privacy preservation promise. In RAPFDL, parties utilize differentially private GAN (DPGAN) to release artificial private data samples confidentially for reliability evaluations rather than share the model parameters or original data. Then, the shared parameter updates are encrypted by the enhanced additively homomorphic encryption algorithm to prevent privacy disclosure during the federated learning training process. Considering that one of our focuses is on distributing different versions of the federated model variants to parties according to their contributions. Especially, fairness is quantified via the correlation coefficient between the final model accuracy and contributions by different parties. We propose the local reliability evaluation mechanism to enhance robustness against poison attacks and enforce fairness for RAPFDL, where each party’s trade model parameter updates with its points. The local reliability and points are initialized in the initial phase and updated via joint model training and local reliability mutual evaluation mechanism.

The fundamental idea is each party is able to earn points by contributing its local model parameter updates to another party. The earned points can be used to trade with other parties. Therefore, each party is motivated to send more private data samples or parameter updates to gain more reward points in the range of the sharing level, and download more parameter updates from the others parties using these points. All the trades are recorded in the blockchain as immutable information, providing auditability and transparency. Particularly, RAPFDL ensures collaborative fairness during upload and download processes as the following:Upload Process: Once receives download request for local parameter updates, the party can determine the number of local parameter updates to send back according to the download request and its own sharing level.Download Process: Since the contribution of each party is different for various parties, the party reliability may be various from the view of different parties. Thus, every party maintains a local reliability list for other parties to record their reliability. The higher the reliability of the party k in the private reliability list of a party, the more likely it is the party will download model parameter updates from it, and thus, more points will be awarded to the party k.

## 5. Implementation of RAPFDL

### 5.1. System Initialization

This section introduces the details of the implementation of the RAPFDL to enhance the efficiency and security of poisoning attacks. The two-stage implementation includes initialization of local reliability values, reward points, and sharing levels, an update of the local reliability lists and reward points in federated model training, and the quantification of fairness.

#### 5.1.1. Initial Evaluation Algorithm

The goal of the initial evaluation algorithm is to access the local data quality of the party through evaluation on artificial private data samples instead of the raw data before federated learning training. Full algorithm flow is as the following: each party trains the DPGAN using local data to generate private data samples, where no true sensitive examples, as well as the distribution of data, will be disclosed, but only some implicit density estimation used in DPGAN. Each party releases its individual generated artificial private data samples according to the local sharing level. The other parties generate the prediction for the artificial private data samples via its local model and return predicted results to parties that released these artificial data samples. There are two prime objectives for sharing artificial DPGAN samples, which are as follows:To gain prior information about the performance of models before federated learning. If the training data of the party is not enough to generate an excellent model, the party will perform poorly in the evaluation algorithm. Thus, other parties shall be more cautious when sharing parameters updates with it, taking into account the evaluation results.To gain the preliminary estimate of the data distribution. Only when the data distribution is different but there is some overlap, can two parties mutually benefit. Assume that two parties $P_{1}$ and $P_{2}$ have released artificial samples with high similarity, which means its local data distributions are nearly the same. Under the circumstances, parameter updates from $P_{2}$ are less likely to improve the model accuracy performance of $P_{1}$. Thus, $P_{1}$ and $P_{2}$ shall avoid sharing parameter updates from each other during the subsequent model training. And other parties should not download the model parameter updates from both $P_{1}$ and $P_{2}$, but from either. Contrarily, assume that two parties have published completely different data distribution artificial samples. Hence, the parameter updates from $P_{2}$ have a negligible enhancement effect on the accuracy of the model of $P_{1}$. Furthermore, suppose that the data distribution of party $P_{1}$ differs from that of all the other parties. Therefore, it is reasonable for other parties to assign low reliability to $P_{1}$ and avoid downloading parameter updates from $P_{1}$ based on previous assumptions. Algorithm 1 presents the detailed procedures of initial evaluation, including local reliability evaluation, sharing level, and reward points initialization and differentially private data samples generation.

**Algorithm 1** Anomalous Model Parameter Detection01. **Server Executes**:02. **the global model parameter initialization**: $w_{0}$
03. **for** iteration round t=1 to I **do**04.    $C_{t}\leftarrow$ Randomly choose N clients05.    **for** client n in $C_{t}$ **do**06.          $L_{t+1}^{n}\leftarrow ClientUpdate(n,w_{t})$
07.      **end for**08.      **Client Detection**09.       **for** each client performing the detection task **do**10.                 $r_{k}\leftarrow$ Return the evaluation results matrix11.      **end for**12.      **for** i=1 to s **do**13.           Calculate the penalty coefficient $f_{t+1}^{i}$
14.      **end for**15.         $w_{t+1}\leftarrow \frac{1}{s}\sum_{i=0}^{s}f_{t+1}^{i}w_{t+1}^{i}$
16. **end for**

#### 5.1.2. Local Reliability Initialization

As to local reliability evaluation, the majority votes of all combined labels are used to compare with predicted labels of one specific party to evaluate the reliability of that party, which takes into account the fact that the predicted labels of the specific party just reflect its outcome, while the majority votes of the combined labels reflect the outcome of majority parties. For instance, party $P_{1}$ releases the DPGAN artificial data samples to the other parties. The other parties label the artificial data samples utilizing its local model, and return predicted labels to party $P_{1}$ for updating the local reliability list. In addition, the party $P_{1}$ labels the data samples via the local model and combines all labels as the cross-validation matrix, in which every column corresponds to the predicted labels of a party. The local reliability of the party j can be initialized as $f_{i}^{j}=\frac{s_{i}}{u_{j}}$, in which $s_{i}$ corresponds to the number of DPGAN artificial private data samples from party i, and $u_{j}$ denotes the number of matches between majority voting labels and predicted labels of party j. In addition, $f_{i}^{j}$ will be normalized in the range of 0 to 1. Once the local reliability of one party is validated to be lower than the threshold $f_{th}$ by the majority of the parties, the party will be banned from all reliability lists owing to potentially low contribution. In RAPFDL, $f_{th}$ is utilized to detect the anomalous party. The low-contribution parties may bypass the detection and access federated learning if the $f_{th}$ is too small. Conversely, most parties may be banned from federated learning. In RAPFDL, a party is likely to download less, even ignoring model parameter updates released by less credible parties, while downloading more those from credible participants.

#### 5.1.3. Sharing Level and Reward Points Initialization

Sharing level corresponds to the upper bound of the artificial private data samples the party can share (i.e., the higher the sharing level is, the more data samples one party can share). According to the number of private data samples $s_{i}$ that party i broadcasts before model training, the sharing level can be initialized as $\delta_{i}=\frac{s_{i}}{L_{i}}$, in which $L_{i}$ is the size of the local dataset. Thus, reward points of party i can be denoted as:(3)$$r_{i}=\delta_{i}*\left|w_{i}\right|*\left(N-1\right)$$
in which $\delta_{i}$ is the sharing level, N is the number of parties, and $\left|w_{i}\right|$ is the number of model parameter updates. The reward points obtained from the initial evaluation can pay for downloading model parameters during the subsequent federated learning training process. The number of model parameters that allow download depends on both the sharing level and local reliability of the requested party.

### 5.2. Differentially Private Data Samples Generation

Although every party just publishes a few unlabeled data samples, it might disclose local data privacy during the initial evaluation phase. To solve the problem, this paper proposes a novel algorithm for generating data samples with a generative adversarial network under differential privacy. Under RAPFDL, the Differentially Private GAN (DPGAN) is trained via adding custom noise to model parameter updates during GAN learning.

Due to the discriminator of GAN is the only component that can access local private data. Thus, it is reasonable to only train the discriminator of GAN with differential privacy, which guarantees the entire GAN since the generator computations are simply post-processing of the discriminator. Its core consideration follows the post-processing feature of differential privacy technology.

To solve the scalability and stability problem of DPGAN training, multi-fold optimization strategies are adopted, including clustering, warm starting, and weight adaptive clipping, which can significantly improve training utility and stability [17,18]. A differential privacy generator is able to generate an infinite number of private data samples and rigorously guarantee differential privacy of them for the intended analysis. We illustrate DPGAN in an enhanced WGAN framework [19] without loss of generality and let every party generate 2000 artificial private data samples. As presented in the research work [20,21], DPGAN is capable of synthesizing RGB and grey images close to artificial private data samples which are generated via the classic GAN without privacy protection.

In addition, the greater amount of data would cause less privacy loss, enabling more iterations in the range of a moderate privacy budget [22,23]. Data augmentation technology is utilized to expand the data size of parties to 100 times and enables DPGAN to generate more artificial data samples in the range of a moderate privacy budget. Particularly, we augment original data with a height and width shift range of 0.01, and a rotation range of 1. In this paper, we apply moments accountant presented in the research work [24] to solve the privacy spent during model training. It should be noted that every party is able to train the DPGAN and generate artificial private data samples offline individually without affecting federated learning.

### 5.3. Anti Poisoning Privacy-Preserving Federated Learning

The details of the anti poisoning privacy-preserving federated learning are shown in Algorithm 2, including how to preserve privacy during model parameter updates via homomorphic encryption followed by local reliability update, how to update the reward points based on per upload/download, and reliable parties set maintenance via blockchains. Particularly, the download budget for model parameter updates of party i, namely $b_{i}$, is relevant to reward points $r_{i}$ of party i at every round. To be specific, $b_{i}$ should not exceed $r_{i}$, because reward points are not enough for downloading model parameter updates from other parties. Furthermore, $b_{i}$ is able to be assigned according to the existing reward points $r_{i}$ dynamically. For the sake of simplicity, we initialize $b_{i}$ as $r_{i}$ at every round. However, the number of model parameter updates allowed to download depends on sharing levels of requested parties and the local reliability of the requester. Next, we will focus on key technical details about homomorphic encryption, model parameter updates, and local reliability updates.
**Algorithm 2** Anti Poisoning Privacy-Perserving Federated Learning01. **Input**: $C,r_{i},r_{j},s_{i},\delta_{j},w_{i},\Delta w_{i}$.02. **Output**: updated reward points $r_{j}^{\prime },r_{i}^{\prime }$, parameters $w_{i}^{\prime }$, and local reliability $f_{i}^{j^{\prime }}$.03. **1: Trade gradients via sharing level, reward points and local reliability:** At every04. round, the goal of party i is to download $s_{i}=r_{i}$ model parameter updates from05. the other parties, while party j∈C is able to provide about $\delta_{j}*|w_{j}|$ model06. parameter updates, one reward point is spent for every download and rewarded for07. every upload. Parties update their model according to the model parameter updates08. of party j∈C as the following:09. **for** j∈C **do**10.      $s_{i}^{j}=\min\left(f_{i}^{j}*s_{i},\delta_{j}*\left|△w_{j}\right|\right),r_{j}^{\prime }=r_{j}+s_{i}^{j},r_{i}^{\prime }=r_{i}-s_{i}^{j},\Delta w_{j}^{i}=\Delta w_{j}$, party j
11.      first choose $s_{i}^{j}$ meaningful gradients from $\Delta w_{j}^{i}$ according to largest values12.      criterion: sort gradients in $\Delta w_{j}^{i}$ and choose top $s_{i}^{j}$ of them, and mask the13.      remaining $\left|△w_{j}^{i}\right|-s_{i}^{j}$ model parameter updates with 0 as $△{\tilde{w}}_{j}^{i}$
14. **end for**15. **2: Model parameter update:** party i utilizes the secret key $sk_{i}$ to decrypt received16. encrypted symmetric key as $f_{s}k$, and utilizes it to decrypt the encrypted parameter17. updates as $c=Enc\left({\tilde{w}}_{j}^{i},k_{j}\right)$ at the end decrypts the sum of model paramter updates18. via homomorphic encryption and thus local model can be updated via integrating all19. the plain paramter updates ${\tilde{w}}_{i}$ as $w_{i}^{\prime }=w_{i}+\Delta w_{i}+Dec\left(\sum_{j\in C\backslash i}Enc\left(△{\tilde{w}}_{j}^{i},k_{j}\right),-k_{i}\right)=$
20. $w_{i}+△w_{i}+\sum_{j\in C\backslash i}△{\tilde{w}}_{j}^{i}$.21. **3: Local reliability update:** party i publishes $s_{i}$ artificial private data samples to22. other party j for labeling. Mutual evaluation is utilized to compute the local23. reliability of the party j as $f_{i}^{j^{\prime }}$ at current round. Thus party i updates party j
24. local reliability via integrating the historical reliability as $f_{i}^{j^{\prime }}=0.3*f_{i}^{j}+0.7*f_{i}^{j^{\prime }}$.25. **4: Local reliability normalization:**
$f_{i}^{j^{\prime }}=\frac{f_{i}^{j^{\prime }}}{\sum_{j\in C}f_{i}^{j^{\prime }}}$
26. **if** $f_{i}^{j^{\prime }}<f_{th}$ **then**27.     party i will report party j as the party with low contribution.28. **end if**29. **5: Set of reliable party:** The reliable party set in blockchain will be reconstructed in30. form of removing the low-contribution party reported by the majority of parties.

#### 5.3.1. Federated Learning Model Training with Homomorphic Encryption

Sharing model parameter updates of federated learning prevent direct exposure of raw data. However, it might disclose sensitive information indirectly. In order to prevent the exposure of privacy during federated learning, we adopt homomorphic encryption in which parties involved in federated learning can only decrypt the weighted sum of parameters rather than single encrypted model parameter updates. To be specific, the Vernam cipher is mathematically proven to be completely secure given sufficient ciphertext and time. Thus, we implement additive homomorphic encryption via provably secure Vernam cipher to achieve secure aggregation of the encrypted data [25,26]. The basic idea for Vernam cipher to form ciphertexts is to combine plaintext with keystream. The security depends on the essential properties: (1) all the operation are modulo of the large integer M; (2) the keystream vary from message to message.

If $p=\max\left(x_{i}\right)$, M is denoted as M=2d∗ln(p∗n). Unless otherwise stated, all subsequent computations are modulo M. The floating-point numbers should be mapped to the field of integer through the SRU algorithm [27]. The pseudo-random function (PRF) can generate pseudorandom keystream via the secure stream cipher, which is similar to Trivium [28,29], keyed with keystream $k_{i}$ of each party and the unique message ID. The secret keys need to be pre-computed via a trusted setting, which can be implemented via the standard SMC protocol or trusted dealers.

For instance, trusted key management authorities are able to generate the keystream, but the generated keystreams can not be used more than once in each communication round. The trusted setting generates $\sum k_{i}=0$, such that every party gains the keystream $k_{i}$. It should be noted that once the blockchain system removes the party i, the set of reliable party C must be reconstructed.

The model parameters are locally updated with gradients encrypted SGD at party i as the following:(4)$$w_{i}{}^{\prime }=w_{i}+\Delta w_{i}+\sum_{j\in C}\Delta {\tilde{w}}_{i}^{j}$$
where $w_{i}$ is model parameters at the current round of party i, $\Delta {\tilde{w}}_{i}^{j}$ is encrypted model parameter updates shared to the party i by the party j.

#### 5.3.2. Local Reliability Update

During every round of federated learning, each party releases a subset of DPGAN artificial private data samples randomly based on its sharing level and evaluates the local reliability of the other parties according to their returned predicted labels, which are calculated via the novel model with local data. The mutual cross-validation mechanism follows the step in Algorithm 1. Eventually, following the same step 4 of Algorithm 2, the local reliability list can be updated via integrating the historical reliability. The local reliability list can be adaptively updated in this way, enabling one to reflect more accurately how much a party contributes to the others parties in federated learning.

### 5.4. Dynamic Asynchronous Federated Learning

In this paper, we propose Dynamic Asynchronous Federated Learning (FedDAsync) considering the averaging frequency control and parameter selection for federated learning to speed up model averaging, which effectively reduces version gap and maintains scheduling flexibility. It enables federated learning to adaptively remove the stragglers with low computing power and bad channel condition. We present the technical details of FedDAsync and explain the key design idea, as shown in Algorithm 3.

A dynamic time window is introduced to control the averaging frequency and reduce the version gap. The central server will wait for the parameters from clients for the next time units (i.e., the window size) for the next aggregation after a complete global model update. All parameters received during the window are utilized to aggregate the global model. The KMeans algorithm is used to set the size of the time window, which clusters the training time of clients to predict the high concurrency of parameter upload. In addition, the weight of data volume is proposed to correct the weight of the version gap, which is determined by the ratio of local data to total data.
**Algorithm 3** Dynamic Asynchronous Federated Averaging (FedDasync)01. **Server Process:**02. **Input:**
α∈(0,1)
03. **Initialize the global model:**
$w_{0},\alpha_{t}\leftarrow \alpha ,\beta_{k}\leftarrow \frac{D_{k}}{D}$
04. **Scheduler Thread:**05. Scheduler periodically triggers some training tasks on some clients, and sents them06. from the latest global model with time stamp.07. **Updater Thread:**08. **for** each round t=1,2… **do**09.     θ=KMeans(timeList,K)
10.     **loop** for θ dynamic seconds after receiving update11.     **Receive** the pair $\left(x_{new},\tau \right)$ from any client12.     timeList.append (t−τ)
13.     $\gamma_{t}\leftarrow \alpha \times S(t-\tau )\times \beta_{k},S(\cdot )$ is function of stateness14.     $x_{t}=\left(1-\gamma_{t}\right)x_{t-1}+\gamma_{t}x_{new}$
15.     **end loop**16.     $x_{t}\leftarrow \sum_{k=1}^{K}\frac{n}{n_{k}}x_{t}^{k}$
17. **end for**

### 5.5. Quantification of Federated Learning Fairness

The fairness of federated learning should be quantified from the aspect of the entire system [30]. We quantify federated learning fairness via the correlation coefficient between the rewards (i.e., different final model accuracies performance) and contribution of a party in this paper (i.e., sharing level reflecting the willingness of parties to share parameter updates, and local model accuracy representing the learning capability on local data).

In particular, we take the contribution as the *x*-axis characterizing the contribution of the parties involved in the federated learning. In settings 1 and 3, we characterize the contribution of the party via local model accuracies performance, because the party with better generalization local data contributes more empirically. In setting 2, we denote the contribution of parties via local model accuracies and sharing levels, because the party whose local data with better generalization and the party who is less private usually contributes more. Furthermore, the party empirically yields higher model accuracy with larger local data in IID scenarios in setting 3. In brief, the *x*-axis is denoted as the Equations (5) and (6), in which δ is the sharing level and u is the local model accuracy:(5)$$x=\{un_{1},\ldots ,un_{n}\}setting\;1\&3$$
(6)$$x=\{\{\frac{\delta_{1}}{\sum \delta_{j}},\ldots ,\frac{\delta_{n}}{\sum \delta_{j}}\}+\{\frac{u_{1}}{\sum u_{j}},\ldots ,\frac{u_{n}}{\sum u_{j}}\}\}setting\;2$$

Similarly, we take rewards of parties (i.e., different final model accuracies performance) as the *y*-axis, as denoted via the Equation (7), in which u is the final model accuracy:(7)$$y=\{u_{1},\ldots ,u_{n}\}$$

Since the *y*-axis evaluates the final model accuracy of different parties in federated learning, which is expected to be correlated with the *x*-axis positively to provide better fairness. Thus, we formally define the quantification of collaborative fairness via Equation (8):(8)$$\begin{array}{l} r_{xy}=\frac{\sum_{i=1}^{n}\left(x_{i}-\overset{-}{x})(y_{i}-\overset{-}{y}\right)}{(n-1)s_{x}s_{y}} \end{array}$$
in which $s_{x}$ and $s_{y}$ are the correction of deviation, and $\bar{x}$ and $\bar{y}$ are the means of x and y. Collaborative fairness is within [−1, 1], and higher values imply better fairness. On the contrary, lower values imply poorer fairness.

## 6. Experimental Evaluation

This section evaluates the RAPFDL framework performance via implementing a federated learning prototype and comparing it with the state of art on three benchmark datasets: (1) MNIST; (2) Fashion-MNIST (F-MNIST); and (3) CIFAR-10. In our experiment evaluation, these datasets are utilized to verify the effectiveness of the anomalous parameter updates detection algorithm to poisoning attacks.

### 6.1. Datasets

We implemented experiments under three typical real datasets, which are MNIST, Fashion-MNIST, and CIFAR-10. A brief summary of these datasets is illustrated in Table 1. The MNIST dataset is used for handwritten digit recognition including 60,000 training images and 10,000 test images. The grayscale of these images has been normalized to 28 × 28 pixels. Just like MNIST, the Fashion-MNIST dataset consists of ten classes of images, including fashion items such as trousers, sandals, bags, et al. All 70,000 examples are a 28 × 28 gray-level image. The CIFAR-10 dataset contains 60,000 images with 32 × 32 color pixies (50,000 for training and 10,000 for testing) in ten classes of images such as racing cars, birds, and airplanes.

### 6.2. Experiment Setup

We build the federated learning environment with the PyTorch framework, Python version 3.6.4, and Numpy version 1.14.0. All experiments are evaluated on the centos7 server with the NVidia P4000 GPU with 32GB RAM. For neural network models of experiments, we construct a CNN model with a softmax output layer for the MNIST and F-MNIST task, two convolutional layers with 32 and 64 channels respectively, and a fully connected layer with 512 neurons. In total, it has about 1.6 million parameters. For the CIFAR-10 task, the network structure is a lightweight ResNet18 model, which includes 17 convolution layers and a fully-connected layer and has about 2.7 million parameters in total. The number of parties for MNIST, F-MNIST, and CIFAR-10 datasets is 12, 12, and 30 respectively. All benign clients train locally using the GradientDescentOptimizer with default initial learning rate and batch size of 64. Each experiment runs 200 communication rounds of federated learning. The scenarios of poisoning attacks include the single attacker and multiple attackers, where one or more parties are assumed as the attacker while the rest of the parties are benign. For the MNIST dataset, the label flipping attack is implemented via changing all the labels from 1 to 5 on the data of malicious clients. For F-MNIST and CIFAR-10 datasets, we mislabel 2% samples with the normal class in the same way as [2]. The backdoor attack is implemented like the prior work [3]. Examples of racing cars are relabeled as birds. The dataset & exp setup details are provided in Table 2. 

To enhance fairness, each party trains 20 epochs individually before federated learning training. The local reliability threshold is set as $c_{threshold}=\frac{1}{|N|-1}*\frac{2}{3}$ through grid search empirically in all experiments, where |N| is the number of reliable parties. Furthermore, we evaluate three settings as the following. Setting 1: Same data size, same sharing level: in this situation, sharing level is set as 0.1 for each party, i.e., parties publish 10% meaningful parameter updates during federated learning. Setting 2: Same data size, different sharing level: sharing level is sampled from [0.2, 0.5) for each party randomly, and parties publish meaningful parameter updates as per private sharing level during federated learning. Setting 3: Different data sizes, same sharing level: we simulate this case where different parties have the same sharing level but different data sizes. Particularly, we partition total {5000, 10,000, 20,000, 30,000} examples among {2, 4, 8, 12} parties respectively for MNIST dataset. The sharing level is fixed to 0.1 for each party. It should be noted that our Setting 2 and Setting 3 are relatively conservative. RAPFDL can result in higher fairness via increasing the contribution diversity in the parties, for instance, sampling sharing level from [0, 1] rather than [0, 0.5] and partitioning data size in an imbalanced way among parties.

### 6.3. Experimental Results

Learning efficiency: The visualized performance is shown in Figure 2, FedDAsync outperforms other baselines with the most rapid learning curves and reduces the training time by 30%. As for the impacts of data heterogeneity, FedDAsync consistently performs well when the data distributions are highly heterogeneous, as shown in Figure 3. Due to the advantages induced from dynamic time window and weights correction, the gain of FedDAsync is more significant in the case of data heterogeneity.

The accuracy over rounds. We demonstrate that the more party involved in federated learning the higher accuracy of the model. We implemented 1, 2, 3, 4, 5, 6, 7, 8, 9, 10, 11, 12 parties involved in the federated learning separately. Each party trains its model with the local dataset containing 5000 images. It is obvious that the more the party participates, the larger the size of the total dataset (i.e., the size of the total dataset is 5000 × 12 for E12). We also implemented a baseline party that only trains the local model on its dataset without federated learning. The training accuracy in E12 is shown in Figure 4, which illustrates that the training accuracy in federated training is higher than the accuracy obtained by the baseline party.

Evaluating the penalizing coefficients. Once poisoned submodels are detected to be anomalous, they will be penalized by the coefficient rather than excluded in aggregation. Figure 5 presents the influence of the penalizing coefficient in aggregation. Assuming that there are 9 normal submodels and a submodel completes the poisoning attack, the attack will be considered successful when the poisoned submodel has not been reported. If any client reports the poisoned submodel and the penalizing coefficient α are assigned to 0.75, the poisoning attack will still work. When the penalizing coefficient is assigned to 0.5, the subtask accuracy is about 15%, meaning the poisoning attack suffers a low success rate. It is reasonable to initial the penalizing coefficient value as 0.5. In this case, the submodel still can improve the global model accuracy even it is misclassified by clients.

Defending against multiple attackers. Figure 6 illustrates the performance of the four approaches defending multiple poisoners: a 1–7 poisoning attack is performed on an original federated learning system namely baseline, Multi-Krum, FoolsGold, and our proposed solution CtrbEval. Figure 6 presents the effectiveness of CtrbEval against FEDAVG which is the fundamental federated learning aggregation framework, where clients locally perform multiple iterations of training before sharing parameter updates with the central server.

## 7. Conclusions

This paper proposes the federated learning framework with the robustness of the poisoning attack and fairness considerations. We explore the poisoning attack and present a novel local reliability mutual evaluation mechanism to effectively detect anomalous updates without directly inspecting raw data. Rather than equipping servers, we propose focusing on delegating the anomaly detection task to clients to cross-validation among parties whose local data can help to evaluate the model performance. In addition, the approach of differential privacy is integrated into the design to guarantee privacy during the evaluation process. In addition, our design can achieve collaborative fairness, by introducing the local reliability, sharing level, and transaction points. The experimental results demonstrated that our scheme is robust to the existing various poisoning attacks and achieves reasonably good fairness, confirming the applicability of our proposed scheme.

## Figures and Tables

**Figure 1 sensors-22-00684-f001:**
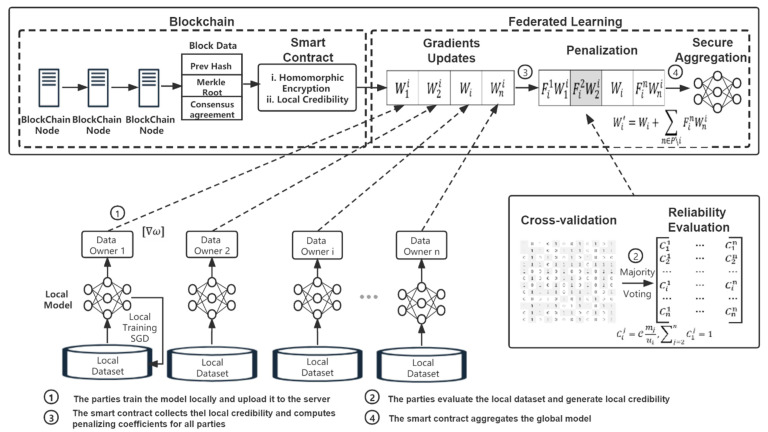
The system overview.

**Figure 2 sensors-22-00684-f002:**
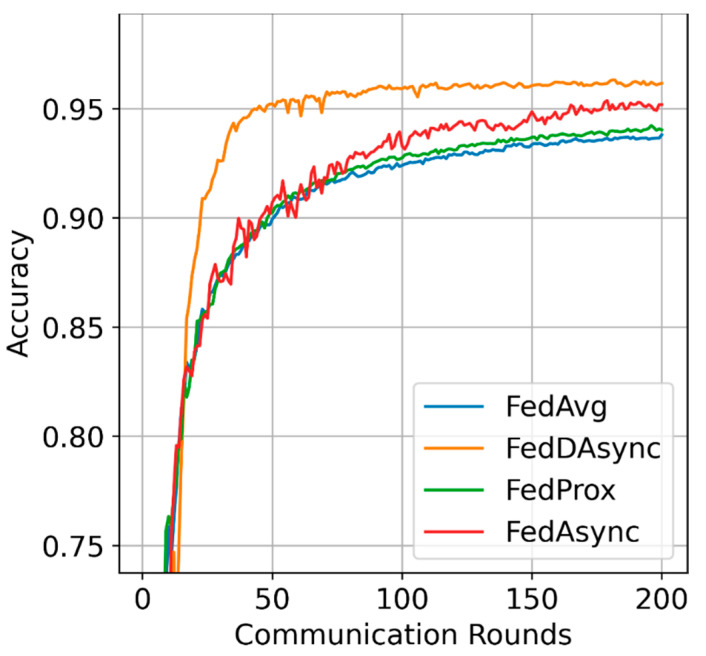
Visualized performance.

**Figure 3 sensors-22-00684-f003:**
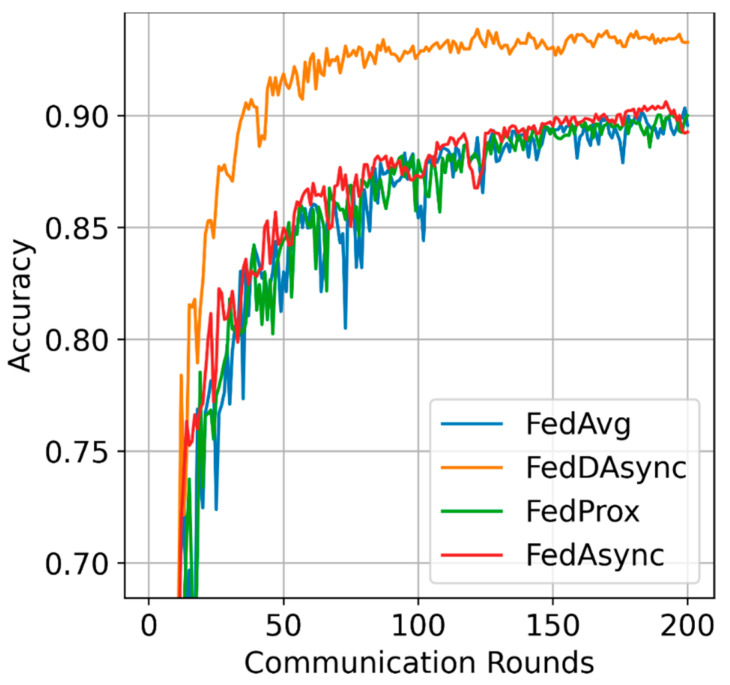
Visualized performance in case of data heterogeneity.

**Figure 4 sensors-22-00684-f004:**
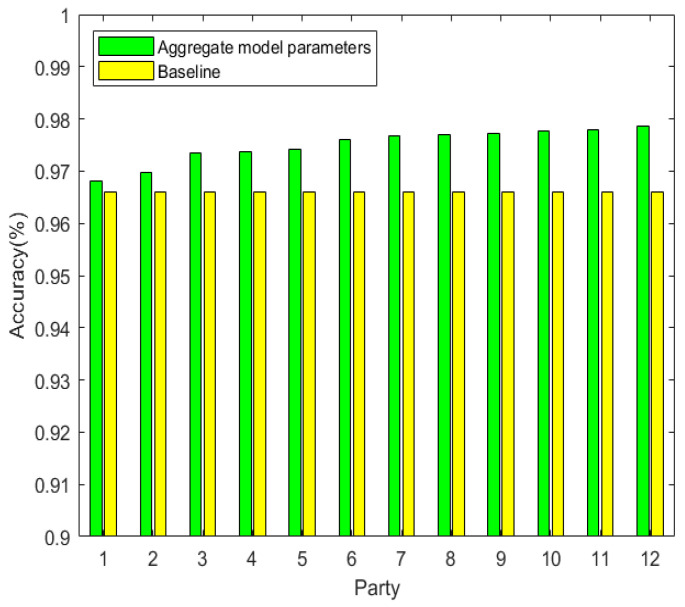
The accuracy over rounds.

**Figure 5 sensors-22-00684-f005:**
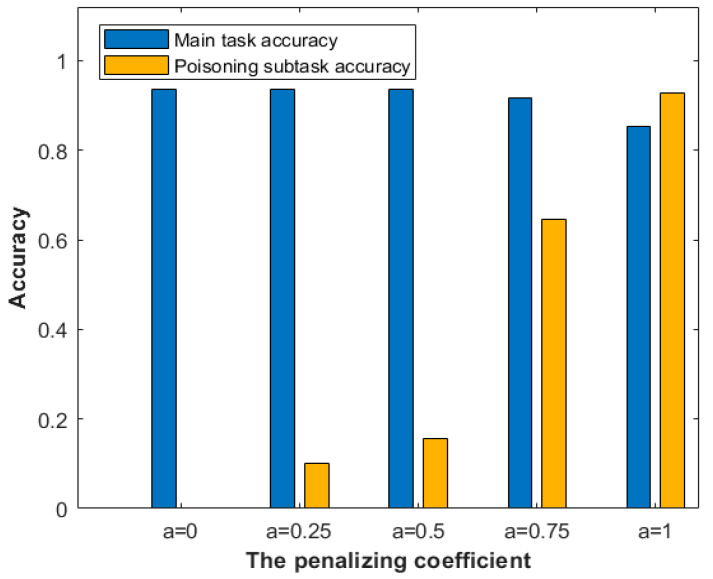
The accuracy of the global model vs. the penalized coefficient.

**Figure 6 sensors-22-00684-f006:**
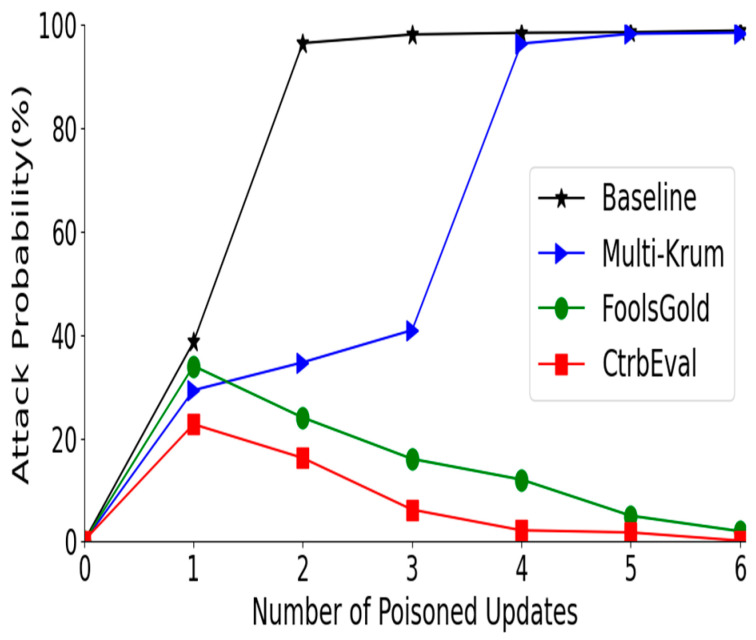
The performance of approaches defending multiple attackers.

**Table 1 sensors-22-00684-t001:** Comparison of three types of blockchain.

Property	Public Blockchain	Consortium Blockchain	Private Blockchain
Read Permission	Public	Restricted	Restricted
Immutability	Nearly impossible	Could be tampered	Could be tampered
Efficiency	Low	High	High
Centralized	No	Partial	Yes

**Table 2 sensors-22-00684-t002:** Summary of datasets & exp setup used in our experiments.

Dataset	Input Size	Training Samples	Testing Samples	Structure
MNIST	28 × 28 × 1	60,000	10,000	CNN
F-MNIST	28 × 28 × 1	60,000	10,000	CNN
CIFAR-10	32 × 32 × 1	50,000	10,000	ResNet18

## Data Availability

Not applicable.

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
