# Peer review of "Dynamic Asynchronous Anti Poisoning Federated Deep Learning with Blockchain-Based Reputation-Aware Solutions"

_sensors, 2022, doi:10.3390/s22020684_

Round 1
Reviewer 1 Report
Sufficient discussion of related work is missing, authors are advised to discuss some recent related works and discuss the research gap & underlying shortcomings with existing methods.
Highlight & discuss the motivation of your work.
Mention the contributions of the work in bulleted form.
In algorithm1, step 4: why there is need to randomly choose N clients, is there any heuristics to decide optimal clients. Also, which random generation technique is adopted for experimentation in this algorithm.
In Algorithm 2, step 6, what does it mean by meaningful gradients? The step numbering if this algorithm seems improper.
In page 10, what is p? here, make it clear. Use variable naming properly to diminish ny possible ambiguity.
Present the dataset & exp setup information/details in Tabular form.
Include the performance comparison study with some existing state of the art & related methods.
Fix the existing grammatical errors. Some sentences are short & while some are quite long. Do necessary english editing.
Reviewer 2 Report
This paper proposes a thin dynamic asynchronous algorithm for federated learning that takes into account averaging frequency control and parameter selection to speed up model averaging and improve efficiency. It allows federated learning to adaptively remove stragglers with low computing power, poor channel conditions, or anomalous parameters. To increase the safety of poisoning attacks, a unique local reliability mutual evaluation method is provided, which allows federated learning to recognize the anomalous parameter of poisoning assaults and modify the weight percentage of the model aggregation dependent on the evaluation score.
The article is quite long and presents some quite complicated technical aspects, not understandable for everyone. It also requires a fairly careful reading. This article contains various information, being well structured in 6 chapters (Introduction, Background, The RAPFDL Framework, Implementation of RAPFDL, Experimental Evaluation , Conclusions) each of them having subchapters which describe most in detail what the title suggests. It is clearly written in a scientific style, explaining a lot of information that can be accessed only if the reader has a strong background in mathematics.
In this article, it is proposed a light-weight dynamic asynchronous algorithm for federated learning that takes into account averaging frequency control and parameter selection. This allows federated learning to adaptively remove stragglers with low computing power, bad channel conditions, or anomalous parameters. Furthermore, a novel local reliability mutual evaluation mechanism is presented to improve the security of poisoning attacks. This mechanism allows federated learning to detect the anomalous parameter of poisoning attacks and adjust the weight proportion of in model aggregation based on the evaluation score. The experimental results revealed that their method is resistant to different poisoning attempts and provides a reasonable level of fairness, indicating that they suggested scheme is applicable.
More details about the blockchain technology and experimental testbed should be provided. Also, more related work about distributed ledger technologies and rationale for choosing a specific blockchain technology for use cases should be provided, for example:
- Palm, Emanuel, et al. "Ricardian Contracts for Industry 4.0 via the Arrowhead Contract Proxy." 30th International Symposium on Industrial Electronics (ISIE), June 20-23, 2021, Kyoto, Japan.. 2021.
- Nadrag Carmen, et al. "Comparative analysis of distributed ledger technologies." 2018 Global Wireless Summit (GWS). IEEE, 2018.
- Alcaraz, Cristina, Juan E. Rubio, and Javier Lopez. "Blockchain-assisted access for federated smart grid domains: Coupling and features." Journal of Parallel and Distributed Computing 144 (2020): 124-135.
This article is detailed written in a scientific paper style and I find it hard to be understood by novice users. Starting from the title, it is suggested to be read and analyzed by someone with a set of knowledge in the Federated Deep-Learning / Blockchain domains. Both the abstract and the conclusion have a correlation to the article's content, figures and tables and are connected to one another, from hypothesis to the results..
Formatting can be improved:
- adjust the images from page 8-9, since the text is hard to understand (make it straight)
- add spaces between tables/figures and text ( 1.0 Space between paragraphs )
Round 2
Reviewer 2 Report
Most of the comments have been addressed, but references to related work for practical use cases in critical infrastructures, especially smart grid, are missing, for example:
- Alcaraz, Cristina, Juan E. Rubio, and Javier Lopez. "Blockchain-assisted access for federated smart grid domains: Coupling and features." Journal of Parallel and Distributed Computing 144 (2020): 124-135.
- Sachian, Mari-Anais, et al. "SealedGRID: Secure and Interoperable Platform for Smart GRID Applications." Sensors 21.16 (2021): 5448.
